# Optimising Interventions for Catheter-Associated Urinary Tract Infections (CAUTI) in Primary, Secondary and Care Home Settings

**DOI:** 10.3390/antibiotics9070419

**Published:** 2020-07-17

**Authors:** Marta Wanat, Aleksandra J. Borek, Lou Atkins, Anna Sallis, Diane Ashiru-Oredope, Elizabeth Beech, Christopher C. Butler, Tim Chadborn, Susan Hopkins, Leah Jones, Cliodna A. M. McNulty, Nia Roberts, Karen Shaw, Esther Taborn, Sarah Tonkin-Crine

**Affiliations:** 1Nuffield Department of Primary Care Health Sciences, University of Oxford, Oxford OX2 6GG, UK; aleksandra.borek@phc.ox.ac.uk (A.J.B.); christopher.butler@phc.ox.ac.uk (C.C.B.); sarah.tonkin-crine@phc.ox.ac.uk (S.T.-C.); 2Centre for Behaviour Change, University College London, London WC1E 6BT, UK; l.atkins@ucl.ac.uk; 3Public Health England Behavioural Insights, London SE1 8UG, UK; anna.sallis@phe.gov.uk (A.S.); tim.chadborn@phe.gov.uk (T.C.); 4Public Health England, London SE1 8UG, UK; diane.ashiru-oredope@phe.gov.uk (D.A.-O.); susan.hopkins@phe.gov.uk (S.H.); Karen.Shaw@phe.gov.uk (K.S.); 5NHS England and NHS Improvement, London SE1 6LH, UK; elizabeth.beech@nhs.net (E.B.); esther.taborn@nhs.net (E.T.); 6Primary Care and Interventions Unit, Public Health England, Gloucester GL1 1DQ, UK; leah.jones@phe.gov.uk (L.J.); Cliodna.McNulty@phe.gov.uk (C.A.M.M.); 7Bodleian Health Care Libraries, University of Oxford, Oxford OX3 7LF, UK; nia.roberts@bodleian.ox.ac.uk; 8University College London Hospitals, London NW1 2PG, UK; 9NHS East Kent Clinical Commissioning Groups, Canterbury CT1 1YW, UK; 10NIHR Health Protection Research Unit in Healthcare Associated Infections and Antimicrobial Resistance, University of Oxford in Partnership with Public Health England, Oxford OX1 2JD, UK

**Keywords:** catheter-associated urinary tract infections, qualitative, behavioural analysis, rapid review, theoretical domains framework, behaviour change techniques, behaviour change wheel

## Abstract

Catheter-associated urinary tract infections (CAUTI) are common yet preventable. Healthcare professional behaviours, such as reducing unnecessary catheter use, are key for preventing CAUTI. Previous research has focused on identifying gaps in the national response to CAUTI in multiple settings in England. This study aimed to identify how national interventions could be optimised. We conducted a multi-method study comprising: a rapid review of research on interventions to reduce CAUTI; a behavioural analysis of effective research interventions compared to national interventions; and a stakeholder focus group and survey to identify the most promising options for optimising interventions. We identified 37 effective research interventions, mostly conducted in United States secondary care. A behavioural analysis of these interventions identified 39 intervention components as possible ways to optimise national interventions. Seven intervention components were prioritised by stakeholders. These included: checklists for discharge/admission to wards; information for patients and relatives about the pros/cons of catheters; setting and profession specific guidelines; standardised nationwide computer-based documentation; promotion of alternatives to catheter use; CAUTI champions; and bladder scanners. By combining research evidence, behavioural analysis and stakeholder feedback, we identified how national interventions to reduce CAUTI could be improved. The seven prioritised components should be considered for future implementation.

## 1. Introduction

Urinary tract infections (UTIs) are among the most common healthcare-associated infections in the UK. It is estimated that 50% of hospital associated infections are linked to use of catheters [1] leading to increased morbidity, mortality and healthcare costs [2]. Catheter-associated urinary tract infections (CAUTI) are preventable by reducing unnecessary catheter use, length of catheter use, and improving insertion technique [3]. Healthcare professionals (HCPs) can play a key role in reducing CAUTI and interventions which target their behaviour are crucial in delivering optimal patient care [4].

Previous work sought to identify gaps in the national response to CAUTI in primary/community care, secondary care and care homes in England. It identified barriers and facilitators to CAUTI-related behaviours for HCPs; identified 11 nationally-adopted interventions; and established the extent to which barriers and facilitators to CAUTI-related behaviours were targeted by these 11 interventions (shown in Box 1) [5].

Box 1List of 11 previously identified national interventions.
The Health and Social Care Act 2008 Code of Practice on the prevention and control of infections and related guidance (HSC Act 2008)—Guidelines for primary, secondary and community settingNICE QS90: Urinary Tract Infections in Adults (NICE QS90)—Guidelines for community settingNICE QS61: Infection prevention and control (NICE QS61)—Guidelines for primary, secondary and community settingNICE catheter audit tools—tools to help implementation of the clinical guidelines for primary and community settingDepartment of Health and Public Health England (2013): Prevention and control of infections in care homes: an informative resource (DH PHE 2013)—Guidelines for care homesSafety thermometer—a tool for HCP in primary, secondary, community and care homes settingEpic 3—guidelines for preventing healthcare-associated infections in acute settingsHigh Impact Intervention for best practice insertion and care (High Impact)—intervention tools for community and secondary care settingCatheter Care: Royal College of Nursing Guidance for nurses (Catheter Care)—guidelines for nurses in primary, community, secondary and nursing homes settingHOUDINI Protocol (HOUDINI)—a nurse-led catheter removal protocol for secondary careCatheter Passport—document to be completed by both patients and HCPs to consistently manage and remove catheters for primary, community and nursing homes setting


The current study aimed to build on this work and explore how national interventions could be improved. Specifically asking: (1)Which interventions, targeting HCP behaviours, are effective at reducing incidence of CAUTI?(2)What is the content of interventions shown to be effective in research studies in comparison to national interventions?(3)To what extent are key influences on HCP behaviour addressed by effective interventions in comparison to national interventions?(4)How can we better address key influences on HCP behaviour?

The overview of the components of the study is provided in Figure 1.

## 2. Results

### 2.1. Rapid Review 

We identified 37 relevant studies which met the inclusion criteria (Appendix A) [6,7,8,9,10,11,12,13,14,15,16,17,18,19,20,21,22,23,24,25,26,27,28,29,30,31,32,33,34,35,36,37,38,39,40,41,42]. Thirty-five studies were conducted in secondary care (hospital care). Of these, 32 were conducted in the US, two in the UK and one in Spain. Two studies were conducted in US nursing homes. No studies were conducted in primary or community care (Appendix A). All interventions contained multiple components, mainly focused on education (*n* = 37) or training of healthcare professionals (*n* = 22). In 27 interventions, this was complemented by HCPs being provided with feedback on catheter-related knowledge and skills. Twenty-five interventions also involved provision of catheter alternatives such as bladder scanners; or introduced clinical champions to decrease catheter use.

### 2.2. Behavioural Analysis of Intervention Content

We used the Behaviour Change Wheel (BCW) [4,43], Theoretical Domains Framework (TDF) [44] and the Behaviour Change Technique Taxonomy (BCTTv1) [45] to describe the behavioural content of interventions. Previous work conducted the same analyses for the 11 national interventions which allowed comparison between national and research interventions [5].

Research interventions used a wide range of BCTs (27 unique BCTs across 37 interventions). These BCTs targeted eight TDF domains, six of which had been identified as key influences on HCPs’ behaviour (Knowledge; Environmental Context and Resources; Memory, Attention and Decision Making; Social Influences; Social Professional Role and Identity; Beliefs about Consequences). Five Research interventions used 7 out of 9 intervention functions (Education, Enablement, Environmental restructuring, Training, Persuasion, Incentivisation and Modelling) and 3 out of 7 policy categories (Service provision; Guidelines; and Communication marketing). The majority focused on behaviours related to pre-insertion (*n* = 30) and removal of catheters (*n* = 28). 

In comparison to the 11 national interventions (based on previous work) [5], research interventions were delivered through more BCTs (*n* = 24 national vs. *n* = 27 research) and intervention functions (*n* = 4 national vs. *n* = 7 research) but the same number of policy categories (*n* = 3); however, proportionally, national interventions use a good range. Interventions differed in the type of BCTs, intervention functions and policy categories that were used. Table 1, Table 2 summarises intervention functions, policy categories and 10 most frequent BCTs across national and research interventions. 

#### Intervention Content Matched against Key Influences on Behaviour

Research interventions targeted key influences on behaviour using more than 60% of the paired BCTs for 4 out of 6 key TDF domains: Environmental Context and Resources (66% of paired BCTs), Social Influences (70% of paired BCTs), Memory, Attention and Decision Making (75% of paired BCTs) and Social Professional Role and Identity (100% of paired BCTs). For the remaining two domains, research interventions used less than 60% of paired BCTs (Beliefs about Consequences; and Knowledge).

Overall, both national and research interventions to some extent addressed all six key TDF domains. However, the six key TDF domains were targeted to a greater extent in the 37 research interventions than in the 11 national interventions, although there were more research interventions reviewed. Particularly, research interventions addressed 4 out of 6 key TDF domains better than national interventions; the only domain which used over 60% of theoretically congruent BCTs in national interventions was Memory, Attention and Decision-Making (Appendix A).

### 2.3. Stakeholder Feedback 

We identified 39 potential intervention components for the six key TDF domains, representing key influences on HCP behaviours. The list of intervention components comprised suggestions for both stand-alone interventions and intervention components, which could be added within national interventions.

Twenty stakeholders were invited to the focus group and 12 (70%) attended. Two more stakeholders provided feedback by telephone. Stakeholders represented the three settings: secondary care (*n* = 7); primary/community care (*n* = 4); and care homes (*n* = 3). Stakeholders suggested amendments to the proposed intervention components and indicated ones which could be omitted. Based on their feedback, the list of intervention components was further refined by the research team, and then by the project steering group. This resulted in a final list of 20 (Appendix A). All intervention components targeted the six key TDF domains, and the vast majority contained BCTs paired with TDF domains to address potential gaps within national interventions. 

### 2.4. Survey

Of the 23 stakeholders invited, 14 (60%) completed the survey. Participants had expertise in secondary care (*n* = 5), primary/community care (*n* = 5) and care homes (*n* = 6). Two non-respondents represented secondary care; three represented primary/community care and one represented care homes. Seven out of 20 intervention components met the prioritisation criteria. For primary/community care 3/16 components were prioritised, 6/18 for secondary care and 4/14 for care homes (Table 3). Two intervention components met the prioritisation criteria across all three settings; two components met the prioritisation criteria for two settings; and three met the prioritisation criteria for one setting (Appendix A).

We also assessed how well the prioritised intervention components addressed previously identified barriers. Out of 22 barriers [5], sixteen were addressed by the prioritised intervention components. Six barriers were not addressed—three because intervention components addressing these barriers were not prioritised by stakeholders and three because intervention components addressing these barriers were viewed as unfeasible by stakeholders.

## 3. Discussion

This multi-method approach assessed how national interventions to reduce CAUTI could be optimised by using research evidence, behavioural analysis and stakeholder feedback. The rapid review identified 37 effective research interventions. Compared to national interventions, research interventions used different BCTs and intervention functions and used a broader range of both. They addressed the six key TDF domains to a greater extent than national interventions. Research evidence and the behavioural analysis helped identify additional intervention components to further address the six key TDF domains. These were refined through expert input from stakeholders and the steering group. Seven intervention components met the prioritisation criteria indicating that they considered most feasible to implement.

### 3.1. Implications for Existing CAUTI Interventions

Stakeholders across all settings prioritised the need for a closer collaboration between healthcare professionals at the crucial point of patient transfer between different settings. In secondary care and care home settings (although also highly scored in primary care), stakeholders also highlighted the importance of standardised nationwide documentation, accessible across healthcare sectors, with details of date of catheter insertion, reason for catheterisation, an action plan for review and removal and details of difficult catheterisation (if relevant). These priorities are reflected in the current national policy with NHS Improvement recommending Catheter Passports, Catheter Audit tool and Catheter decision tree as a solution to a consistent, evidence-based system wide approach to catheter assessment and management [46,47]. Catheter Passports which include both clinician and patient section also partially address another component prioritised across all settings, namely the need to work closely with patients and families by ensuring that they are involved in discussions about catheter care, which is often difficult to implement. However, it is important to highlight that the uptake of Catheter Passport is still low. A recent survey in the UK indicated that only 7.1% of nursing and care homes has a Catheter Passport scheme in place and only less than one fifth of people with catheters had Catheter Passport [46].

Another important recommended action by the NHS England was a need to review catheter training and audit [47]. In the current study, the need to address staff beliefs and knowledge about risks of using catheters was particularly highlighted in care homes. This is in line with a recent study showing that training on antimicrobial use in nursing and care homes was available for only 6.8% of staff [46]. It is important to ensure that the staff has access to appropriate training to support optimal care for patients.

Finally, stakeholders in secondary and care home settings also prioritised the need for common catheter guidelines across healthcare settings. This is perhaps not surprising given a number of national guidelines which are for use in one setting only.

### 3.2. Optimising National Interventions 

The prioritised intervention components addressed 16/22 previously reported barriers to HCP CAUTI-related behaviour. These could be considered for incorporating as part of existing national interventions. Incorporating these components may need to be carried out by intervention developers or those familiar with the content and delivery methods for these interventions. It is important to highlight that 9 out of 11 national interventions are guidelines; therefore, it is likely that interventions may be somewhat restricted in the type of BCTs which can be incorporated in their content. Where interventions might have overlapping content, it would be helpful to assess whether any interventions could be discontinued or combined, utilising different relevant BCTs to optimise intervention content and delivery. While some guidelines complement each other, it may be more useful to provide fewer but more comprehensive guidelines which could include BCTS not previously utilized while also investing in supporting healthcare professionals to adhere to guidelines. Finally, policy makers may decide, rather than optimising existing interventions, to develop new interventions incorporating existing components, thus utilising BCTs, intervention functions and policy categories not previously used.

### 3.3. Strengths and Limitations

This study further demonstrated the usefulness of using a behavioural analysis approach to identify the behavioural content of interventions and assess how well these address the barriers to desired behaviour change. A number of studies showed the strengths of such approaches in other contexts [48,49,50]. We built on this theoretical approach by focusing on interventions which had shown to be effective research at changing the target behaviours and consulting expert stakeholders to develop new intervention components. This evidence, theory and expert-based approach means that the prioritised intervention components are likely the best approach to optimising current interventions.

However, the study has some limitations. Firstly, while we combined evidence based on research and theory with stakeholder feedback, at times, we had to make a judgment on which source of evidence should be given priority (e.g., when stakeholders suggested intervention components which did not target key TDF domains). In the current study, we have prioritised the views of stakeholders given that their expertise is specific to CAUTI and that theoretical frameworks apply to any behaviour. In addition, we have not gathered feedback from patients and carers. Secondly, our survey had a moderately high response rate (60%); however, the numbers of non-respondents in all three settings was similar. Thirdly, the barriers discussed here are based on evidence from secondary care and the US and there was limited to no evidence on whether the same influences on behaviour are present in primary/community care and care home settings. The current project partially addresses this limitation by gathering feedback from stakeholders in all three settings.

## 4. Materials and Methods 

This was a mixed methods study consisting of four stages. We have combined research evidence, behavioral analysis and expert feedback in order to identify how existing interventions can be optimized using examples from effective research interventions. Behavioral analysis allows an examination of whether content of the interventions addresses barriers and facilitators to CAUTI behaviors. This complemented with feedback from experts aims to provide a full picture of how existing interventions can be further developed [5,51].

### 4.1. Rapid Review to Identify Effective Research Interventions

We searched the following databases up to October 2018: Medline; EMBASE; PsycINFO; Cochrane Database of Systematic Reviews; Cochrane Central Register of Controlled Trials; CINAHL; and PROSPERO for any systematic review which reviewed studies assessing interventions aimed at changing HCPs’ behaviour to reduce incidence of CAUTI in primary care, secondary care, and/or care home settings. We used the following inclusion criteria for reviews: i) published since 1 January 2014; ii) including studies in high income countries; iii) written in English; iv) reporting on effective interventions (statistically significant results). Since the most recent systematic review identified studies up to 2016, we also searched for primary studies since that date. Search terms were informed by previous work [5], and were reviewed by an information specialist (Appendix A). Titles and abstracts of reviews and primary studies were screened against the inclusion/exclusion criteria by MW with 20% screened by AB. Full texts were obtained for abstracts meeting the inclusion criteria and screened by MW. Throughout the screening process, when there was uncertainty about inclusion, texts were discussed with AB and STC. From all included studies, MW extracted data on study design, setting, target behaviour, description of the intervention, outcome measures and effect on incidence of CAUTI.

### 4.2. Behavioural Analysis of Research Interventions in Comparison to National Interventions 

We used the Behaviour Change Wheel (BCW), Theoretical Domains Framework (TDF) and the 93-item taxonomy of Behaviour Change Techniques (BCTTv1) to describe the behavioural content of interventions. The BCW enables characterisation of interventions using nine intervention functions, i.e., purposes that an intervention may serve; and seven policy categories, i.e., channels through which interventions are implemented [4,48] The TDF is an integrative framework of 14 theoretical domains of influences on behavior [49]. Behaviour change techniques (BCTs) are defined as active ingredients of interventions designed to bring about change [50]. For each research intervention, we extracted data on BCTs, intervention functions, policy categories and TDF domains. We also categorised the target behaviour in relation to stage of catheter care: pre-insertion, insertion, maintenance or removal. Previous work conducted the same analyses for the 11 national interventions which allowed comparison between national and research interventions [5].

### 4.3. Assessing the Behavioural Content of Research Interventions against Key Influences on Healthcare Professional Behaviours 

A previous review identified 22 influences on healthcare professional behaviour related to the prevention of CAUTI [5] Authors mapped these influences to the TDF and identified 6 key TDF domains relevant to CAUTI prevention behaviour (Appendix A).

We used the same 6 key TDF domains and BCT × TDF matrix to assess how well effective research interventions address the barriers and facilitators to CAUTI-related behaviours. To facilitate this, we examined the frequency with which BCTs paired with the key TDF domains were present in research interventions [5]. We then compared research and national interventions in relation to the extent to which they targeted the 6 key TDF domains by using paired BCTs [5].

### 4.4. Stakeholder Feedback Using a Focus Group and Survey

Informed by research interventions, we developed a list of potential intervention components targeting the most frequently reported barriers within each of the six key TDF domains. Stakeholders were identified by the project steering group and included HCPs with expertise in managing CAUTI. Stakeholders were invited to attend a 3 h focus group in London. Participants were presented with a description of the 11 national interventions and the list of potential intervention components which could be used to optimise national interventions. Participants provided feedback on intervention components by making suggestions about adding or removing aspects.

The same stakeholders were invited to complete an electronic survey, sent by email. Responses were anonymous. Participants were asked to provide brief demographic details including their roles, work setting or expertise, and years of relevant experience.

Participants read the revised intervention components and were asked to judge two aspects: (a) whether each was (a) relevant and (b) suitable for implementation in primary care, secondary care and/or care homes. We assessed suitability for implementation using the APEASE criteria: Affordability; Practicability; Effectiveness and cost effectiveness; Acceptability; Side effects and safety; and Equity [4].

This gave us two scores: percentage of stakeholders who deemed each intervention component as relevant; and percentage of maximum possible APEASE score (calculated by taking the denominator for each setting and multiplying by 6 which was the number of APEASE criteria). To be prioritised, the intervention components for each setting had to meet two pre-specified criteria: (a) at least 50% of stakeholders who responded for that setting deemed this intervention component to be relevant; and (b) the intervention component scored at least 60% of the maximum APEASE score. This provided a list of prioritised intervention components for each of the three healthcare settings.

The study was reviewed by the University of Oxford Clinical Trials and Research Governance team and deemed a service development study, and consequently did not require a research ethics review.

## 5. Conclusions

This project explored how national interventions targeted at healthcare professionals to reduce CAUTI could be improved for primary/community care, secondary care and care home settings. By drawing on behavioural theory and tools as well as expert stakeholder views and experiences, we identified seven intervention components which were assessed as relevant and feasible for implementation. These intervention components can be considered when optimising individual national interventions to reduce CAUTI.

## Figures and Tables

**Figure 1 antibiotics-09-00419-f001:**
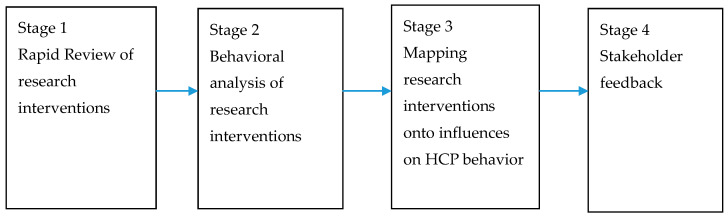
Overview of the study components.

**Table 1 antibiotics-09-00419-t001:** Frequency of intervention functions and policy categories in national and research interventions.

**Intervention Function**	**National Interventions (*n* = 11)**	**Research Interventions (*n* = 37)**
Education	11	37
Enablement	7	31
Training	4	22
Modelling	2	3
Incentivisation	1	5
Environmental restructuring	0	25
Persuasion	0	6
Coercion	0	0
Restriction	0	0
**Policy Category**	**National Interventions (*n* = 11)**	**Research Interventions (*n* = 37)**
Guidelines	9	17
Service provision	1	36
Legislation	1	0
Communication/marketing	0	17
Regulation	0	0
Fiscal measures	0	0
Environmental/social planning	0	0

**Table 2 antibiotics-09-00419-t002:** Ten most frequently identified BCTs in national and research interventions.

National Interventions Top 10 BCTs (*n* = 11)	Frequency	Research Interventions Top 10 BCTs (*n* = 37)	Frequency
Instruction on how to perform a behavior *	10	Instruction on how to perform a behaviour	36
Information about health consequences	9	Feedback on behaviour	27
Self-monitoring of behaviour	5	Adding objects to the environment	22
Social support practical	4	Prompts/cues	15
Information about social environmental consequences	4	Behavioural practice/rehearsal	14
Goal setting behaviour	4	Social support practical	13
Monitoring of behaviour by others without feedback	3	Feedback on outcome of behaviour	13
Feedback on behaviour	3	Action planning	9
Feedback on outcome of behaviour	3	Behavioural substitution	9
Self-monitoring of outcome of behaviour	3	Restructuring the social environment	8

* BCT highlighted in bold Indicates the BCT is commonly used in both national and research interventions.

**Table 3 antibiotics-09-00419-t003:** Overview of prioritised intervention components and the addressed barriers.

Intervention Component	Primary/Community Care	Secondary Care	Care Homes
Creating the rule that staff transferring catheterised patients to another setting, check/review the need for a catheter with the receiving team; could be prompted by a checklist for discharge/admission of patients to another setting*Barriers: Transitions of care; Pre-emptively deciding to insert catheters; Cultural norms regarding standard catheterisation practice for specific patient groups*	+	+	+
Before inserting catheters, staff required to inform patients and relatives about pros and cons of catheters, risks associated with catheter use, including sepsis and antibiotic resistance and the importance of hydration (with or without written resources) and record that this has been explained to patients.*Barriers: Requests from patients and their carers*	+	+	+
Ensure availability of setting and profession specific guidelines which are in agreement and which include examples of how to adapt to local contexts where possible.*Barriers: CAUTI guidelines not perceived as relevant*	+	+	
Standardised nationwide computer-based documentation, accessible across healthcare sectors, requiring person initiating urinary catheterisation to insert detailed information such as date of catheter insertion, reason for catheterisation, an action plan for review and removal and details of difficult catheterisation (if relevant). Provided when transferring patients across settings.*Barriers: inconsistent documentation and records; Transitions of care; No information regarding placement/ duration of catheters*		+	+
Intervention to persuade staff of benefits of not using catheters for both patients (e.g., loss of mobility, bed sores, lower risk of infection) and staff (e.g., fewer patients developing infection, improved patient outcomes, lower costs). Reassure staff that not using catheters does not lead to suboptimal care and reframing severity of CAUTI as patient safety issue with a story of a patient who contracted CAUTI.*Barriers: Convenience and ease of monitoring; Perceived severity of CAUTI; Lack of perceived benefits to CAUTI interventions; Lack of awareness of the risks related to catheters*			+
Introduction of “CAUTI Champions” (nurses and doctors). Champions role model how to manage patient/carer requests for catheter, lead on staff education and provide practical support for colleagues wanting to support patients to TWOC (trial without catheter)*Barriers: Physicians dictate nurses’ practice; Lack of peer support and buy-in*		+	
Provision of bladder scanners, accompanied by staff training on how to use scanners, to aid decisions in relation to problems with urinary retention.*Barriers: Unavailability of medical alternatives to urinary catheterisation; Lack of knowledge of how to manage patients without catheters*		+

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
