# Peer review of "Optimising Interventions for Catheter-Associated Urinary Tract Infections (CAUTI) in Primary, Secondary and Care Home Settings"

_antibiotics, 2020, doi:10.3390/antibiotics9070419_

Round 1

Reviewer 1 Report

Wanat et al. performed a mixed method approach to address behavior components of healthcare workers and patients to introduce interventions that can reduce catheter-associated urinary tract infections(CAUTI). Overall study design and findings are relevant and useful to the field, yet incremental in nature over existing interventions.

Author needs to address some minor concern to highlight the finding better:

  1. Define all terms and short-namenclature in the beginning, as currently those are towards the end of manuscript in the method section e.g. BCT ?
  2. Provide reference to APEASE criteria.
  3. Define Behavior change wheel , behavior change taxonomy and theoretical domain framework after the initial rapid review section. Also, why authors have choosen these methods, should be clear to the readers. It makes no sense to have a rapid review section both in main text and methods but no details of core approach in main text.
  4. Line 257 : responses were anonymous? Then how to know identity of the groups these stakeholder belong to?
  5. In the second intervention form table 3:

  “Before inserting catheters, staff required to inform patients and relatives about pros and cons of catheters, risks associated with catheter use, including sepsis and antibiotic resistance and the importance of hydration (with or without written resources) and record that this has been explained to patients. “

 Isn’t that a standard practice of medicine, if this is prioritized to that level there is a severe problem of the practice and behavior of catheter usage. This needs to be stressed.

Reviewer 2 Report

The paper "Optimising interventions for catheter-associated urinary tract infections in primary, secondary and care home settings" deals with improving health care interventions to reduce the frequency of catheter-associated urinary tract infections. It is designed in part as a literature review to identify which interventions from the Research literature are effective, and in part as a behavioural analysis to describe what behavioural content lies in these measures. Furthermore, the Authors investigated to what degree interventions from the Research literature and national guidelines target six previously reported Central aspects of catheter-associated UTI prevention behaviuour. Finally, a Group of clinical experts from different levels in the Health care system were consulted to produce a prioritized short-list of seven suggested interventions.

In essence, this is a service Development study. At the same time, however, it is also a complex effort to dig deeper into the matter of Healthcare provider behaviour, based both on current Research literature and expert opinion, and address the most Central behavioural items that come to play in preventing CAUTIs. I think that the paper fits well in this journal, but could equally be published in behavioural science or nursing medicine journals.

The biggest strength of the paper is that it looks into the behavioural content of current recommendations, and Explores in a structured manner how well they address core behavioural impediments to infection prevention. It does so in a theory-driven and qualitative manner. As such, the paper could serve as a model for how to improve adherence to guidelines in any field of medicine. It is clearly written, With understandable Language, and appropriate use of tables and boxes for clarity. The introduction is very Clear and concise. Also, Table 2 is very easy to read and understand at a glance.

Its main weakness is that it does not test in a quantitative manner whether its short-list of seven interventions Works better to prevent infections than any other short-list of interventions. In this reviewer's opinion, however, this would be the topic of an entirely New study, and should not act as an argument against the present paper.

I think that the Discussion section could be improved:

-From the numbers, it seems that stakeholders were recruited in equal part from secondary care, primary care and care homes. But they represented only 60% of the stakeholders invited. It would be interesting in the Discussion part to mention at what care level the non-responders worked, and say something about the representation of stakeholders here. To this reader, it seems that secondary care is relatively under-represented, and it would strengthen the Discussion if this was addressed.

- Should we not strive for fewer guidelines? The paper points to eleven national intervention guidelines. And it does a mammoth work of comparing these guidelines to up to date behavioural research. The paper does not discuss, however, whether or not one should strive for fewer guidelines, or more, for that matter. This reader thinks fewer guidelines would be a natural way to og.

Minor weaknesses: - I miss a Figure 1 outlining the four different steps on the way towards the final short-list. The four different steps would be literature review, behavioural analysis of Research interventions, mapping of Research interventions onto influences on Healthcare provider behaviour, and the stakeholder feedback. This reader had to read the article several times to understand the many steps undertaken here.

- Line 71: Secondary care. To this Reader, who does not work in the UK, it is not immediately obvious what is meant by secondary care. It would be easier if you clearly state here that both general practice, hospitals, and care homes are included.

- Line 80: The abbreviations BCT and TDF should be written out the first time they are used. (They are written out in the M&M section, which unfortunately comes after the Results section.)

- Table 2: Do you only use a finite number of predefined categories to describe national and research interventions, and are the categories the same «on both sides» (both national and research sides)? It does not say clearly in the M&M section.

- Line 125: This sentence is difficult to understand, and should be reformulated for clarity.

- Line 182: Starting with «Given», this sentence is difficult to understand, and should be reformulated for clarity.

- Conclusion: I think the conclusion is good, and well-written.

Round 2

Reviewer 1 Report

I am satisfied by the author's responses.